# Occurrence of Ten Protozoan Enteric Pathogens in Three Non-Human Primate Populations

**DOI:** 10.3390/pathogens10030280

**Published:** 2021-03-02

**Authors:** Estelle Menu, Bernard Davoust, Oleg Mediannikov, Jean Akiana, Baptiste Mulot, Georges Diatta, Anthony Levasseur, Stéphane Ranque, Didier Raoult, Fadi Bittar

**Affiliations:** 1Department of Epidemiology of Parasitic Diseases, Aix Marseille University, IRD, AP-HM, SSA, VITROME, 13005 Marseille, France; Stephane.ranque@ap-hm.fr; 2IHU Méditerranée infection, 13005 Marseille, France; bernard.davoust@mediterranee-infection.com (B.D.); oleg.mediannikov@ird.fr (O.M.); georges.diatta@ird.fr (G.D.); anthony.levasseur@univ-amu.fr (A.L.); Didier.raoult@ap-hm.fr (D.R.); fadi.bittar@univ-amu.fr (F.B.); 3Department of Epidemiology of Parasitic Diseases, Aix Marseille University, IRD, AP-HM, MEPHI, 13005 Marseille, France; 4Laboratoire National de Santé Publique, Brazzaville BP 120, Congo; jakiana2000@yahoo.fr; 5ZooParc of Beauval, Zoological Research Center, 41110 Saint-Aignan-sur-Cher, France; baptiste.mulot@zoobeauval.com; 6Campus International UCAD-IRD, Université Cheikh Anta Diop de Dakar, Dakar 18524, Senegal

**Keywords:** non-human primate, protozoa, qPCR

## Abstract

Non-human primate populations act as potential reservoirs for human pathogens, including viruses, bacteria and parasites, which can lead to zoonotic infections. Furthermore, intestinal microorganisms may be pathogenic organisms to both non-human primates and humans. It is, therefore, essential to study the prevalence of these infectious agents in captive and wild non-human primates. This study aimed at showing the prevalence of the most frequently encountered human enteric protozoa in non-human primate populations based on qPCR detection. The three populations studied were common chimpanzees (*Pan troglodytes*) in Senegal and gorillas (*Gorilla gorilla*) in the Republic of the Congo and in the Beauval Zoo (France). *Blastocystis* spp. were mainly found, with an occurrence close to 100%, followed by *Balantidium*
*coli* (23.7%), *Giardia*
*intestinalis* (7.9%), *Encephalitozoon*
*intestinalis* (1.3%) and *Dientamoeba*
*fragilis* (0.2%). None of the following protozoa were detected: *Entamoeba*
*histolytica*, *Enterocytozoon*
*bieneusi*, *Cryptosporidium*
*parvum*, *C*. *hominis*, *Cyclospora*
*cayetanensis* or *Cystoisospora*
*belli*. As chimpanzees and gorillas are genetically close to humans, it is important to monitor them frequently against different pathogens to protect these endangered species and to assess potential zoonotic transmissions to humans.

## 1. Introduction

Several enteric protozoa that can provoke zoonotic diseases are currently facilitated by and associated with the growing popularity of open farms and petting zoos, which are a possible source of outbreaks and transmission to humans [1,2,3,4]. The parasitic ecology of non-human primates varies depending on the wild or captive nature of the species studied [5]. Although several intestinal protozoan species have been involved in diseases affecting captive non-human primates [6], data on the pathogenicity of parasites in non-human primates are scarce. A few studies have reported that enteric protozoan pathogens, namely *Giardia* spp., *Entamoeba histolytica* and *Cryptosporium* spp., may cause gastrointestinal enteritis in non-human primates (gibbons, orangutans, marmosets, gorilla and chimpanzees) [7,8].

In recent years, molecular studies have identified common zoonotic parasites, including *Giardia intestinalis*, *Entamoeba histolytica* and *Cryptosporidium* sp. [5], in captive primates but not in the wild, suggesting possible transmission of these pathogens from humans. However, *Cryptosporidium* and *Giardia* have been isolated from wild primates living in disturbed forests, possibly as a result of their close contact with humans [9]. Moreover, most parasites carried by non-human primates pose a potential zoonotic risk to researchers and caretakers in breeding centers [10,11], acting as reservoirs for human infections [12]. It is, therefore, important to document and monitor protozoan parasites that are present in these primates, whether they are wild or captive. Regularly monitoring primates for pathogens, including protozoan parasites, is warranted and may prevent zoonotic transmission to humans and protect these endangered species. This study aimed at investigating the occurrence of the most common human enteric protozoa in non-human primate populations [13], namely *Blastocystis* spp., *Cryptosporidium parvum*, *C. hominis*, *Cyclospora cayetanensis*, *Dientamoeba fragilis*, *Giardia intestinalis*, *Balantidium coli*, *Entamoeba histolytica* and *Cystoisospora belli*, along with two microsporidia, namely *Enterocytozoon bieneusi* and *Encephalitozoon intestinalis*. These pathogens, which have been previously identified as the most frequent human parasites [13], lead to human diseases ranging from chronic intestinal colonization with abdominal discomfort and sporadic diarrhea to severe enteric disorders. In addition, many protozoa are overlooked, neglected or underestimated by traditional microscopic examination, depending on the microscopist’s experience and the sensitivity of the methods used. To overcome this, we employed molecular techniques to diagnose these parasites. Three primate populations were studied, namely common chimpanzees (*Pan troglodytes*) in Senegal and gorillas (*Gorilla gorilla*) in the Republic of the Congo and in the Beauval Zoo (France), both to improve the health and care of these great apes and to estimate the potential risk of zoonotic infection. Moreover, *Blastocystis* diversity (genotypes) was also studied in each primate population. This study was done to provide an update on the prevalence of the most frequently encountered human enteric protozoa in non-human primate populations based on qPCR detection.

## 2. Results

### 2.1. Parasitological Analysis

A total of 76 stool samples from great apes were analyzed using real-time PCR assays targeting ten protozoa. The parasitic infection rate (i.e., detection of at least one parasite per sample) was 97.4% (74/76 samples). The most prevalent protozoan parasite among all tested samples was *Blastocystis*, with a prevalence of 97.4%. Two other pathogenic protozoa also occurred with relatively high prevalence rates, namely *Balantidium coli* (at 23.7%) and *Giardia intestinalis* (at 7.9%). The remaining parasites were either detected at a low prevalence (<2%) or not detected at all.

### 2.2. Prevalence of Parasites by Primate Species and Site

#### 2.2.1. Chimpanzees from Senegal

At least one parasitic infection was detected in 97.9% (47/48 samples) of the 13 individual chimpanzees tested from Senegal (Table 1). *Blastocystis* spp. was the most common protozoan with a prevalence rate of 97.9% (47/48), followed by *Dientamoeba fragilis* with a prevalence rate of 25% (12/48) and *Giardia intestinalis* with a prevalence rate of 2.1% (1/48) (Table 1).

#### 2.2.2. Gorillas from the Republic of the Congo

All gorilla samples collected from the Republic of the Congo were positive for at least one of the ten tested parasites (19/19, 100%) (Table 1). *Blastocystis* was reported in all tested samples, followed by *Balantidium coli* in 11 samples (57.9%), *Giardia intestinalis* in five samples (26.3%) and *Encephalitozoon intestinalis* in one sample (5.3%).

#### 2.2.3. Gorillas from the Beauval Zoo

Of the nine stool samples collected from nine different gorillas in the Beauval Zoo, at least one of the ten tested parasites was detected in 8 samples (88.9%). In the same way, as the wild great apes from Senegal and the Republic of the Congo, *Blastocystis* spp. was the most common protozoan, with a prevalence of 88.9% (eight positive samples) (Table 1). The *Balantidium coli* infection rate was 77.8% (7/8 samples). No other parasite was detected.

### 2.3. Genotyping of Blastocystis spp.

Of the 74 samples positive for *Blastocystis*, 19 samples were from gorillas (the Republic of the Congo), 47 samples were from chimpanzees (Senegal) and eight samples were from gorillas (Beauval Zoo, France); three subtypes (STs) were detected (Table 2). ST1 was the most abundant (*n* = 60, 81.1%), followed by ST2 (*n* = 10, 13.5%) and ST5 (*n* = 4; 5.4%) (Table 2). A phylogenetic analysis was generated to illustrate the relationship between the different subtypes obtained in our study, along with other *Blastocystis* ST1, ST2, ST3 and ST5 sequences recovered from human, chimpanzee and gorilla species, which are available in the GenBank database (Figure 1). This figure shows that sequences from each ST lineage were clustered together as a monophyletic group and supported by bootstrap values higher than 50%. However, paraphyletic and polyphyletic clades resulting from certain nucleotide variations between sequences (bootstrap values below 50%) were formed within each ST branch (Figure 1).

## 3. Discussion

Our survey revealed a high rate of *Balantidium coli* infection in gorillas from the Republic of the Congo and captive gorillas from France, at 57.9% and 77.8%, respectively (Table 1). *B. coli* is a cosmopolitan ciliate that colonizes the intestines of many animals, including humans, with pigs serving as reservoir hosts [6,14]. In captive gorillas, infections with *B. coli* are usually without clinical signs. Although in some cases a mild or self-limiting disease may result upon first exposure to this protozoan, life-threatening *B. coli* infections have also occurred in gorilla populations [15,16].

*Dientamoeba fragilis* was found only in the chimpanzees from Senegal, with a prevalence of 25% (Table 1). This flagellate protozoan parasite has been reported in great apes [17]. Its pathogenicity has been demonstrated in Lowland gorillas from Cameroon [18].

We also found *Giardia intestinalis* in wild primates, with a prevalence of 2.1% in chimpanzees from Senegal and a much higher prevalence (26.3%) in gorillas from the Republic of the Congo (Table 1). This is in line with numerous previous studies that reported infection by this enteric protozoan in wild non-human primates and more frequently in captive ones [19,20,21,22,23]. It has previously been shown that the occurrence of *Giardia* in some wild primate populations is associated with a synanthropic lifestyle [9].

*Encephalitozoon intestinalis* was detected in one gorilla from the Republic of the Congo (Table 1). It is known that microsporidia can infect apes, as it was demonstrated through experimental infections [24], while *E. intestinalis* has previously been found in wild non-human primates [25,26].

The PCR-based diagnostic method used in our study highlighted a *Blastocystis* sp. prevalence of 97.4% in the great apes studied (Table 1). This ubiquitous microeukaryote is a commensal of the large intestine of more than one billion people in both developed and developing countries [27]. These data are consistent with previous studies showing that non-human primates are common hosts of this protozoon eukaryote [5], regardless of their lifestyle and geographical origin [28]. In the current study, *Blastocystis* ST1 occurred in 100% of chimpanzees from Senegal (Table 2; Figure 1). ST1 was also the major subtype found in gorillas, with a prevalence of 48.2% (13/27). Various epidemiological studies conducted in primates strongly suggest that the predominant subtypes, ST1, ST2 and ST3, are shared by captive and wild monkeys, and thus naturally occur in this host group [28,29,30]. While ST1 is the most frequent subtype in non-human primates, it is the second most common subtype in humans [28,29,30]. Finally, we found an additional subtype, ST5, in captive gorillas from France as compared to wild gorillas from the Republic of the Congo (Table 2; Figure 1). ST5 has previously been found in Western Lowland gorillas in French zoos [28]. One explanation might be that they are in close contact with other animals in captivity and also have direct and repeated contact with humans. In fact, the Beauval Zoo has the largest diversity of captive animals in France, with 600 species claimed. It is, therefore, not surprising that ST5 was only found in the gorillas from the zoo. Other subtypes not found in this study have been previously described in non-human primates, such as ST8 [30] and ST10-17 [31]. It is notable that ST8 has also been found in zookeepers working in close contact with non-human primates, highlighting the risk of zoonotic transmission [29]. Finally, our phylogenetic tree (Figure 1) shows the presence of some nucleotide variations between sequences from the same ST. This finding suggests the probable existence of “subgroups or subclades” within each *Blastocystis* ST, which require further investigation (e.g., by using longer small subunit ribosomal DNA gene sequences).

In summary, the present study illuminates our current understanding of parasitosis in ape populations and the occurrence of *Blastocystis* among these species. Furthermore, we think that an observational survey of pathogens is required at each opportunity and that similar studies should be conducted regularly, as the dynamics of microorganisms are not stable and each pathogen can emerge at any moment, leading to an epidemic situation and possible intra- or interspecies transmission.

## 4. Materials and Methods

### 4.1. Sample Collection and Information

Stool samples were collected from primates living in the Republic of the Congo, Senegal and at the Beauval Zoo (Saint-Aignan-sur-Cher, France). Nineteen gorilla samples were collected at the Lesio-Louna Lefini Gorilla Nature Reserve, located at 2°58′033.1″ S and 15°28′033.4″ E in the Republic of the Congo. Nine samples were collected at the Beauval Zoo, located at 47°14′51″ N and 1°21′12″ E, France, in 2015. Forty-eight chimpanzee stools were collected at the Dindefelo community natural reserve at 12°23′00″ N and 12°19′00″ W, Senegal, in 2016. Each collected sample was identified, placed in a 50-millilitre tube, stored in dry ice and then transported to the laboratory, where it was stored at −80 °C. All samples were collected around feeding sites or night nests, in line with standard practice.

### 4.2. Ethical Statement

No experimentation was conducted on the animals. All stool samples were collected from the soil. In the Republic of the Congo, the access permit to the Lesio-Louna Lefini Gorilla Nature Reserve was delivered by the Forestry Economy and Sustainable Development Ministry (0094/MEFDD/CAB/DGACFAP-DTS de la Direction Générale de l’Agence Congolaise de la Faune et Aires Protégées, on 24 August 2015). The permit to collect samples was delivered by the Environment and Sustainable Development Ministry of Senegal for chimpanzee stool samples (001914/DEF/DGF de la Direction des Eaux, Forêts, Chasses et de la Conservation des Sols, on 5 June 2016). No other permits were required, as no invasive procedure was used in this study and the collection of the samples did not disturb the wild fauna.

### 4.3. DNA Extraction from Stool Samples

DNA extraction was performed using the EZ1 (Qiagen GmbH, Hilden, Germany) automated protocol with pretreatment [14]. The extraction protocol was adapted for stool processing as follows: 200 mg of stool sample was added to 350 µL of G2 lysis buffer (Qiagen GmbH, Hilden, Germany) in a tube containing glass powder and then disrupted in a FastPrep-24 grinder (MP Biomedicals) at maximum power for 40 s. After 10 min of incubation at 100 °C to allow complete lysis, the tubes were centrifuged at 10,000 *g* for one minute. Then, 200 µL of supernatant was added to a tube containing 20 µL of Proteinase K, which was incubated overnight at 56 °C. 

To control for DNA extraction quality and the absence of PCR inhibitors, universal eubacterial primers and probes were used to amplify 16S rRNA bacterial genes, and a qPCR technique named “all-bacteria” (TTB) was performed on each extracted DNA sample. Positive result indicated the absence of PCR inhibitors, while negative results indicated that the DNA extraction was repeated.

### 4.4. Singleplex Real-Time PCR Amplification and Detection

Ten different specific primers and TaqMan^TM^ hydrolysis probes targeting different sequence regions were used in multiparallel assays. All primers and probe sequences used in this study have previously been published and used [32,33] (Table 3). The real-time PCR reactions were conducted using 20 µL total volumes containing 10 µL of Master mix (Roche Diagnostics GmbH, Mannheim, Germany), 0.5 µL of each primer, 0.5 µL of probes, 3 µL of distilled water, 0.5 µL of UDG and 5 µL of DNA. Analyses were performed using a CFX96^TM^ Real-Time PCR detection system (BIO-RAD, Life Science, Marnes-la-Coquette, France). Amplification reactions were performed as follows: two minutes of incubation at 50 °C, five minutes incubation at 95 °C, followed by 40 cycles of five seconds at 95 °C and 30 s at 60 °C. Real-time PCR results were considered negative when the Ct value was more than 38 or no amplification curve was obtained, as described in previous studies [32,34].

### 4.5. Amplification of the Small Subunit Ribosomal DNA (SSU rDNA) Gene and Blastocystis Typing 

For each *Blastocystis*-positive sample, the extracted DNA was subjected to a standard polymerase chain reaction (PCR) assay using the *Blastocystis*-specific primers BL18SPPF1 (5′-AGTAGTCATACGCTCGTCTCAAA-3′) and BL18SR2PP (5′-TCTTCGTT ACCCGTTACTGC-3′) designed by Poirier et al. [44]. These primers target a 320 to 342 bp DNA fragment located in the *Blastocystis* sp. SSU rDNA gene. Amplification was performed in a total volume of 50 μL with AmpliTaq Gold^®^ 360 (Thermo Fisher Scientific, Waltham, MA, USA). After denaturation at 95 °C for 15 min, 40 cycles of amplification were performed with a 2720 Thermal Cycler^TM^ (Applied Biosystems, Courtaboeuf, France) as follows: 30 s at 95 °C, 30 s at 59 °C and 30 s at 72 °C. The amplification products were analyzed by electrophoresis using 1.5% weight-per-volume (*w*/*v*) agarose gel containing SYBR^TM^ Safe DNA Gel Stain (ThermoFisher). The PCR products were purified using MultiScreen^®^ PCR (Merck Millipore, Darmstadt, Germany) and sequencing reactions were carried out by using a DNA sequencing kit (BigDye Terminator Cycle Sequencing v1.1 Ready Reactions; PE Biosystems) according to the manufacturer’s instructions. Sequencing products were purified and electrophoresis was performed with a 3130 Genetic Analyzer (Applied Biosystems). The nucleotide sequences were analyzed with the CodonCode Aligner software (www.codoncode.com/aligner (accessed on 10 January 2021)) and compared with those available in the GenBank database by using the BLASTn programme (www.ncbi.nlm.nih.gov/BLAST (accessed on 10 January 2021)). Subtypes (STs) were then determined by selecting the best match (exact/closest similarity) with all *Blastocystis* sp. subtype sequences available in the GenBank database. Phylogenetic analyses were performed using the maximum likelihood method implemented in MEGA X software (https://www.megasoftware.net/ (accessed on 10 January 2021)). The maximum likelihood analysis was based on the Tamura–Nei model. Bootstrap values were calculated from 1000 replicates.

### 4.6. Accession Numbers

The nucleotide sequences for *Blastocystis* sp. were deposited in GenBank under the accession numbers MT676292 to MT676365.

## 5. Conclusions

Given the potential zoonotic risk of protozoan parasites and their pathogenicity, it is important to monitor their presence in captive and wild animals. Of the ten tested protozoa, five were detected in great apes. Therefore, non-human primates may present a potential reservoir for *Balantidium coli*, *Blastocystis* sp., *Dientamoeba fragilis*, *Encephalitozoon intestinalis* and *Giardia intestinalis*. This study highlights the high prevalence (close to 100%) of *Blastocystis* sp. colonization in both wild and captive primates, which is higher than previously reported. In line with previous studies, ST1 followed by ST2 were the predominant subtypes. *Blastocystis* ST5 was only detected in captive gorillas and might be acquired through close contact with human or other animal species.

## Figures and Tables

**Figure 1 pathogens-10-00280-f001:**
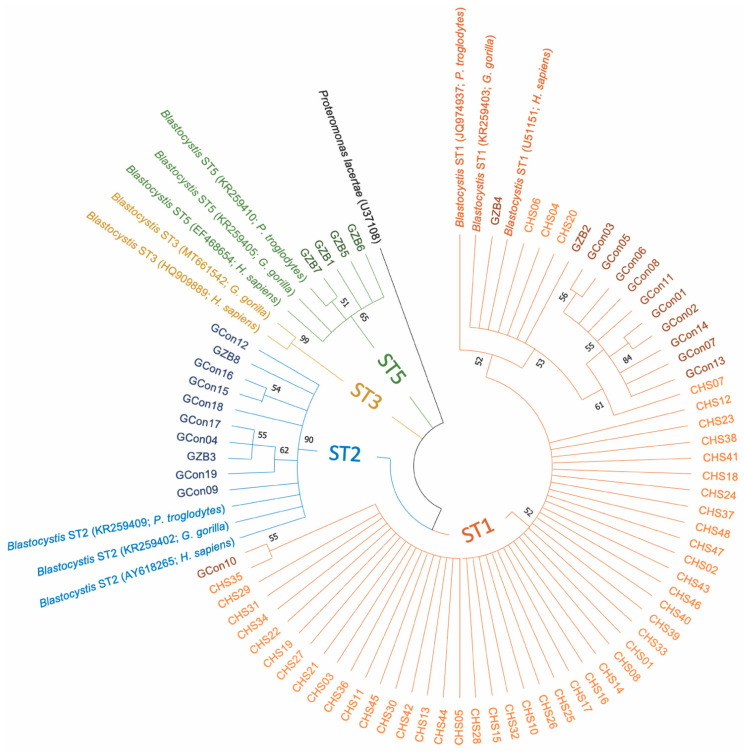
Phylogenetic tree based on the small subunit ribosomal DNA (SSU rDNA) gene sequences, showing the relationships between the different subtypes found in our samples and reference sequences from GenBank. The molecular phylogenetic analysis was carried out using the Maximum Likelihood method based on the Tamura–Nei model in Molecular Evolutionary Genetics Analysis X (MEGA X) software. Bootstrap values (obtained from 1000 replicates) lower than 50% were removed. The branches are displayed with different colors according to the sequence subtype (ST): ST1, orange; ST2, blue; ST3, gold; ST5, green. The sequences obtained in this study from chimpanzees and gorillas are represented by lighter and darker colors than those of the corresponded STs. CHS, chimpanzees from Senegal; GCon, gorillas from the Republic of the Congo; GZB, gorillas from the Beauval Zoo. T5 13.m Beauval zoo, France (%).

**Table 1 pathogens-10-00280-t001:** Prevalence of ten protozoa in the studied non-human primates.

Tested Parasites	Chimpanzees (*n* = 48) from Senegal		Gorillas (*n* = 19) from the Republic of the Congo		Gorillas (*n* = 9) from the Beauval Zoo, France		Overall
	Number	%	Number	%	Number	%	%
*Giardia intestinalis*	1	2.1	5	26.3	0	0	7.9
*Cryptosporidium* spp.	0	0	0	0	0	0	0
*Balantidium coli*	0	0	11	57.9	7	77.8	23.7
*Blastocystis* spp.	47	97.9	19	100	8	88.9	97.4
*Cyclospora cayetanensis*	0	0	0	0	0	0	0
*Dientamoeba fragilis*	12	25.0	0	0	0	0	0.2
*Enterocytozoon bieneusi*	0	0	0	0	0	0	0
*Encephalitozoon intestinalis*	0	0	1	5.3	0	0	1.3
*Entamoeba histolytica*	0	0	0	0	0	0	0
*Cystoisospora belli*	0	0	0	0	0	0	0

**Table 2 pathogens-10-00280-t002:** *Blastocystis* subtypes in the studied non-human primates.

	Positives	ST1		ST2		ST5	
	Number	Number	%	Number	%	Number	%
Chimpanzees from Senegal	47	47	100	0	0	0	0
Gorilla from the Republic of the Congo	19	11	57.9	8	42.1	0	0
Gorilla from the Beauval Zoo, France	8	2	25	2	25	4	50
Total	74	60	81.1	10	13.5	4	5.4

**Table 3 pathogens-10-00280-t003:** List of primers and probe sequences used in this study.

Parasite	Target Gene	Primer/Probe Names	Sequence 5′-3′	Source
		BcoliF	TGCAATGTGAATTGCAGAACC	
*Balantidium coli*	ITS1	BcoliR	TGGTTACGCACACTGAAACAA	[32]
		BcoliP	FAM-CTGGTTTAGCCAGTGCCAGTTGC-TAMRA	
		Blasto FWD F5	GGTCCGGTGAACACTTTGGATTT	
*Blastocystis* sp.	18S	Blasto R F2	CCTACGGAAACCTTGTTACGACTTCA	[35]
		Blasto probe	FAM-CCTACGGAAACCTTGTTACGACTTCA-MGB	
		1PS_F	AACTTTAGCTCCAGTTGAGAAAGTACTC	
*Cryptosporidium hominis/parvum*	Hsp70	1PS_R	CATGGCTCTTTACCGTTAAAGAATTCC	[36]
		Crypt_P	FAM-AATACGTGTAGAACCACCAACCAATACAACATC-TAMRA	
		Cyclo250F	TAGTAACCGAACGGATCGCATT	
*Cyclospora cayetanensis*	18S	Cyclo350R	AATGCCACGTAGGCCAATA	[37]
		Cyclo281T	FAM-CCGGCGATAGATCATTCAAGTTTCTGACC-TAMRA	
		Ib-40F	ATATTCCCTGCAGCATGTCTGTTT	
*Cystoisospora belli*	ITS2	Ib-129R	CCACACGCGTATTCCAGAGA	[38]
		Ib-81Taq	FAM-CAAGTTCTGCTCACGCGCTTCTGG-TAMRA	
		Df-124F	CAACGGATGTCTTGGCTCTTTA	
*Dientamoeba fragilis*	18S	Df-221R	TGCATTCAAAGATCGAACTTATCAC	[39]
		Df-172revT	FAM-CAATTCTAGCCGCTTAT-TAMRA	
		FEI1	GCAAGGGAGGAATGGAACAGAACAG	
*Encephalitozoon intestinalis*	18S	REI1	CACGTTCAGAAGCCCATTACACAGC	[40]
		PEI1	FAM-CGGGCGGCACGCGCACTACGATA-TAMRA	
		Ehf	AACAGTAATAGTTTCTTTGGTTAGTAAAA	
*Entamoeba histolytica*	18S	Ehr	CTTAGAATGTCATTTCTCAATTCAT	[41]
		Ehp	FAM-ATTAGTACAAAATGGCCAATTCATTCA-TAMRA	
		FEB1	CGCTGTAGTTCCTGCAGTAAACTATGCC	
*Enterocytozoon bieneusi*	18S	REB1	CTTGCGAGCGTACTATCCCCAGAG	[42]
		PEB1	FAM-ACGTGGGCGGGAGAAATCTTTAGTGTTCGGG-TAMRA	
		Giardia-80F	GACGGCTCAGGACAACGGTT	
*Giardia intestinalis*	18S	Giardia-127R	TTGCCAGCGGTGTCCG	[43]
		Giardia-105T	FAM-CCCGCGGCGGTCCCTGCTAG-TAMRA	

## Data Availability

Not applicable.

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
