# Peer review of "Occurrence of Ten Protozoan Enteric Pathogens in Three Non-Human Primate Populations"

_pathogens, 2021, doi:10.3390/pathogens10030280_

Round 1
Reviewer 1 Report
The manuscript adds nothing new to the present state of knowledge. The mere statistical compilation of the qPCR results of a small number of samples, without in-depth bioinformatics analysis and the lack of any scientific concept is visible in the work. As the authors themselves mention in a very short discussion, similar studies with similar results have already been published. Therefore, in this study, there is no novelty in the approach to the obtained data. Moreover, the authors focused only on protozoa, which, as a source of zoonoses, do not pose a serious threat to humans. Perhaps the research should also be extended to other groups of pathogens: including viral or bacterial infections.
Author Response
The assessment of zoonotic risk by determining the prevalence/occurrence of different pathogens should not be limited to one or few studies, but it should be conducted regularly as the dynamic of microorganisms is not stable and each pathogen can emerge at any moment leading to an epidemic situation and a possible intra/inter-species transmission. Thus, observational survey of pathogens, such as our study, is required at each opportunity and should not be discouraged by the presence of similar studies/similar results.
Many studies target bacterial and viral pathogens but few focus on protozoan species and/or are limited to a handful of protozoan parasites, while our study systematically targets 10 enteric parasites that are frequently associated with human enteric disorders. Moreover, different studies alert that enteric protozoa contribute to the global burden of infectious diseases and indeed cause severe morbidity and mortality in humans (in addition to animals) worldwide, particularly in developing countries. Of note that such risk might be further potentiated by animal and human movement (immigration), climatic changes (raising temperature) and the increase of water and food contamination upon rapid populations’ growth. Although protozoan parasites are largely neglected during laboratory diagnosis — probably because the lack of sensitivity using microscopic examination — the molecular investigations presented here are the only alternative to estimate the real occurrence of these pathogens.
As pointed out by the Reviewer, the limited number of stool samples results from the difficulty of collecting them from wild and endangered animals such as non-human primates without stressing these protected species and disturbing the wild fauna (of note that only non-invasive samples (feces) can be collected). Although the present study further illuminates our current understanding of Blastocystis occurrence among apes’ populations, the goal of this study is not to draw an exhaustive statistical picture of parasitosis in these species. Hence, the use of bioinformatic tools is out of the scope of the present study.

Reviewer 2 Report
The manuscript by Menu et al. reported a survey conducted on non-human primate (NHP) populations for 10 common enteric protozoan pathogens, not lengthily. The bottom line is, a phylogenetic analysis of the sequenced Blastocystis spp. isolates is indispensable. Also, a table should be provided with all of the primers and probes details information (PCR and qPCR) used in this study. A thorough English check is mandatory (including sentence structures and spelling), it is too hard to follow the writing. Some of the scientific names were not italicized.
Other comments:
Title: Line 1-3: The title should be changed. It describes about “diversity and distribution of eukaryotic enteric pathogens”, while the study focuses on only protozoan agents rather than other eukaryotic enteric pathogens like fungi, nematodes etc. Also it describes only about some common pathogenic protozoa rather than all of them. Also, the “diversity and distribution” part should be corrected. If you name it as “diversity and distribution”, then you should add pyrosequencing of 18S rRNA and do the taxonomic classification of the pathogens. So the title should be: “Occurrence/incidence/prevalence of some common enteric pathogenic protozoa in three non-human primate populations” or something like that.
Abstract: Line 13-23: An incomplete abstract. There should be a conclusive line at the end of the paragraph describing the real importance of this study for public health.
Introduction: Line 26-45: Introduction is also not completed. Need couple more lines at the end describing the importance of the study. Please add one or two more paragraphs about the pathogenic protozoans found in NHPs.
Result: Line 55-74: please re-write the sentences.
Line 70: Add a new section for this result part as 3.3 Genotyping of Blastocystis sp.
Line 75: Don’t put any caption naming “Table”
Line 76 & 77: Table 1 and Table 2: Please add couple more columns to separate the number values from the percent values. It doesn’t look good together. Also, it is confusing.
Discussion: Please provide some more information about the subtypes of NHPs. There are several reports available describing prevalence of other subtypes among NHPs like subtype 8 or subtypes 10-17.
Methods: Line 162: please describe with couple lines about how the subtyping was done after sequencing i.e. by blast searching or what? What is SSU rDNA? Please write the abbreviation of the gene.
Author Response
Review 2
The manuscript by Menu et al. reported a survey conducted on non-human primate (NHP) populations for 10 common enteric protozoan pathogens, not lengthily. The bottom line is, a phylogenetic analysis of the sequenced Blastocystis spp. isolates is indispensable. Also, a table should be provided with all of the primers and probes details information (PCR and qPCR) used in this study. A thorough English check is mandatory (including sentence structures and spelling), it is too hard to follow the writing. Some of the scientific names were not italicized.
R: We thank Reviewer 2 for his/her comments. A table containing all the primers and probes used in this study was added in the revised manuscript. English reviewing was checked by professional English editing service. Scientific names of species were verified and italicized accordingly.
Other comments:
Title: Line 1-3: The title should be changed. It describes about “diversity and distribution of eukaryotic enteric pathogens”, while the study focuses on only protozoan agents rather than other eukaryotic enteric pathogens like fungi, nematodes etc. Also it describes only about some common pathogenic protozoa rather than all of them. Also, the “diversity and distribution” part should be corrected. If you name it as “diversity and distribution”, then you should add pyrosequencing of 18S rRNA and do the taxonomic classification of the pathogens. So the title should be: “Occurrence/incidence/prevalence of some common enteric pathogenic protozoa in three non-human primate populations” or something like that.
R: We agree with Reviewer 2 and we have changed the title as: “Occurrence of ten protozoan enteric pathogens in three non-human primate populations”
Abstract: Line 13-23: An incomplete abstract. There should be a conclusive line at the end of the paragraph describing the real importance of this study for public health.
R: As recommended by Reviewer 2, a conclusive line was added at the end of the abstract (lines: 24-26).
Introduction: Line 26-45: Introduction is also not completed. Need couple more lines at the end describing the importance of the study. Please add one or two more paragraphs about the pathogenic protozoans found in NHPs.
R: As recommended by Reviewer 2, the importance of the study and the pathogenic protozoans found in NHPs were provided in the Introduction section (Line 55).
Result: Line 55-74: please re-write the sentences.
R: This paragraph was re-written in the revised manuscript.
Line 70: Add a new section for this result part as 3.3 Genotyping of Blastocystis sp.
R: We agree with the reviewer and we have added a sub part "2.3. Genotyping of Blastocytis sp." In the Result section.
Line 75: Don’t put any caption naming “Table”
R: We agree with the Reviewer 2 and we have deleted it.
Line 76 & 77: Table 1 and Table 2: Please add couple more columns to separate the number values from the percent values. It doesn’t look good together. Also, it is confusing.
R: We agree with the Reviewer 2 and we have added more columns to separate the number values from the percent values.
Discussion: Please provide some more information about the subtypes of NHPs. There are several reports available describing prevalence of other subtypes among NHPs like subtype 8 or subtypes 10-17.
R: We agree with the Reviewer 2 and we have added this sentence about other subtypes found in NHPs in the discussion section (Line 141).
Methods: Line 162: please describe with couple lines about how the subtyping was done after sequencing i.e. by blast searching or what? What is SSU rDNA? Please write the abbreviation of the gene.
R: As recommended by the Reviewer 2, we have explained how the subtyping was done in the revised manuscript on the “4.5. Amplification of the small subunit ribosomal DNA (SSU rDNA) gene and Blastocystis typing” section. Moreover, we have specified the abbreviation in the title.

Reviewer 3 Report
The manuscript on enteric pathogens in primates by Estelle Menu et al. presents a small but interstimg and maybe important dataset for evaluating the importance of enteric pathogens within primates including humans. I thus recommend this manuscript for publication after thoroughly considering the following suggestions.
Title:
The title could be related better to the text by using 'protozoan enteric pathogens' instead of"eukaryotic enteric pathogens".
Abstract:
Although often read this way, it is discouraged to use "NHP" for "non-human primate". Simple means like avoiding abbreviations render the complete text easier to read and make it more attractive.
Introduction:
The reviewer recommends, shortly to introduce information on effects in the primates mentioned, caused by the pathogens studied here. This would help to make the contribution more attractive for those that are not familiar with all of these protozoa.
It could also be very helpful to provide a short review on the main protozoa found in humans. It is recommended to come back to this point in the Discussion paragraph. This could enable the reader easier and better to evaluate the importance of these pathogens.
Results:
In order to be really helpful for those who are inspired by this paper to perform comparable studies, it is regarded tremendously helpful to include a short table with the utilised primer sequences, their range of use, and the corresponding references. This very simple addition will highly increase the impact of this contribution.
General:
Please check again for italics for genus and species names. The manuscript is not consistent in this respect. Please also check carefully for following journal style.
Author Response
Review 3
The manuscript on enteric pathogens in primates by Estelle Menu et al. presents a small but interstimg and maybe important dataset for evaluating the importance of enteric pathogens within primates including humans. I thus recommend this manuscript for publication after thoroughly considering the following suggestions.
R: We thanks the Reviewer 3 for his/her supportive comment.
Title:
The title could be related better to the text by using 'protozoan enteric pathogens' instead of"eukaryotic enteric pathogens".
R: We agree with the Reviewer 3 and we have changed the title as the following: “Occurrence of ten protozoan enteric pathogens in three non-human primate populations”
Abstract:
Although often read this way, it is discouraged to use "NHP" for "non-human primate". Simple means like avoiding abbreviations render the complete text easier to read and make it more attractive.
R: We agree with the Reviewer 3 and we have changed “NHPs” by “non-human primates” throughout the manuscript.
Introduction:
The reviewer recommends, shortly to introduce information on effects in the primates mentioned, caused by the pathogens studied here. This would help to make the contribution more attractive for those that are not familiar with all of these protozoa.
R: As recommended, the effects of these protozoan in primates (mainly human) were briefly added in the introduction section.
It could also be very helpful to provide a short review on the main protozoa found in humans. It is recommended to come back to this point in the Discussion paragraph. This could enable the reader easier and better to evaluate the importance of these pathogens.
R: We have specified, in the revised manuscript, that the 10 selected protozoa are among the most frequent protozoan parasites found in humans.
Results:
In order to be really helpful for those who are inspired by this paper to perform comparable studies, it is regarded tremendously helpful to include a short table with the utilised primer sequences, their range of use, and the corresponding references. This very simple addition will highly increase the impact of this contribution.
R: As recommended by the Reviewer 3, a table with all primers, probes and the corresponding references was added in the revised manuscript (Table 3).
General:
Please check again for italics for genus and species names. The manuscript is not consistent in this respect. Please also check carefully for following journal style.
R: Italics for genus and species names were verified in addition to the journal style.

Round 2
Reviewer 1 Report
Thank you for responding to the comment. The corrections made by the authors significantly increase the value of the manuscript.
Author Response
Please find enclosed the manuscript entitled “Occurrence of ten protozoan enteric pathogens in three non-human primate populations” by Estelle Menu et al. that we submit for publication as an original article.
We hope that you consider this article suitable for publication in your journal and look forward to your response.
Answer to Review 1
Thank you for responding to the comment. The corrections made by the authors significantly increase the value of the manuscript.
R: We thank Reviewer 1 for his/her comments.
Reviewer 2 Report
The manuscript has been improved to a great extent from its previous version. However, I still don’t see the phylogenetic analysis of the sequenced Blastocystis spp. isolates, which is not hard to do. All of the information are in online and there are several free software programs available to do the job (including MEGA5).
Other minor comments:
Introduction: Please add a conclusive line in your introduction. Like: This study was done to provide information…. Something close to it.
Discussion: A conclusive line for discussion as well. Like: We hope that this research can assists…
Tables: Table 1 is not well fitted with the manuscript. Please adjust the table. For table 3, better if you could fit the table 3 in a single page.
Also, no need to convert the whole manuscript into landscape format. Only table 3 can be in landscape and rest of the manuscript can be in portrait format.
Author Response
Please find enclosed the manuscript entitled “Occurrence of ten protozoan enteric pathogens in three non-human primate populations” by Estelle Menu et al. that we submit for publication as an original article.
We hope that you consider this article suitable for publication in your journal and look forward to your response.
Answer to the Review 2
The manuscript has been improved to a great extent from its previous version. However, I still don’t see the phylogenetic analysis of the sequenced Blastocystis spp. isolates, which is not hard to do. All of the information are in online and there are several free software programs available to do the job (including MEGA5).
R: We did not understand that a phylogenetic tree was required. We have therefore added a phylogenetic tree in this new version as Figure 1. As we have a high number of Blastocystis sequences, we have provided a circular phylogenetic tree. However, if Reviewer 2 prefer a classical tree, we can also provide it in a further revision.
Other minor comments:
Introduction: Please add a conclusive line in your introduction. Like: This study was done to provide information…. Something close to it.
R: This was added as recommended by Reviewer 2.(Line 65-67).
Discussion: A conclusive line for discussion as well. Like: We hope that this research can assists…
R: This was also added in the revised manuscript (Lines: 162).
Tables: Table 1 is not well fitted with the manuscript. Please adjust the table. For table 3, better if you could fit the table 3 in a single page.
R: Tables 1 and 3 were adjusted in the revised manuscript.
Also, no need to convert the whole manuscript into landscape format. Only table 3 can be in landscape and rest of the manuscript can be in portrait format.
R: We have checked, and the manuscript is in portrait format. The margin and appearance of the manuscript appear to have been decided by the publisher. If this format is not suitable, would it be possible to get help from the publisher?

Round 3
Reviewer 2 Report
The manuscript is almost ready to go now. Although, the discussion part is still lacking the citations from the tables and figure. Please cite your data with parentheses accordingly in your discussion section. Also describe a little about your phylogenetic tree in the result section 2.3 and discussion section (like how the sequences clustered together based on their sequence similarities, distances from each other based on PP and bootstrap support values etc.).
Author Response
Review 2
The manuscript is almost ready to go now. Although, the discussion part is still lacking the citations from the tables and figure. Please cite your data with parentheses accordingly in your discussion section. Also describe a little about your phylogenetic tree in the result section 2.3 and discussion section (like how the sequences clustered together based on their sequence similarities, distances from each other based on PP and bootstrap support values etc.).
R: We agree with the reviewer.
We added the citations from tables and figure in the discussion section.
We added “A phylogenetic analysis was generated to illustrate the relationship between the different subtypes obtained in our study, along with other Blastocystis ST1, ST2, ST3 and ST5 sequences recovered from human, chimpanzee and gorilla species which are available in the GenBank database (Figure 1). This figure shows that sequences from each ST lineage were clustered together as a monophyletic group and supported by bootstrap values higher than 50%. However, paraphyletic and polyphyletic clades, resulted from some nucleotide variations between sequences (bootstrap values below 50%), were formed within each ST branch (Figure 1).” In the result section 2.3.
We added “Finally, our phylogenetic tree (Figure 1) showed the presence of some nucleotide variations between sequences from the same ST. This finding suggests the probable existence of “subgroups or subclades” within each Blastocystis ST which requires further investigations (e.g., by using longer SSU rDNA gene sequences).” In the discussion section (line 171).
